# The Role of Perineural Invasion in Prostate Cancer and Its Prognostic Significance

**DOI:** 10.3390/cancers14174065

**Published:** 2022-08-23

**Authors:** Yuequn Niu, Sarah Förster, Michael Muders

**Affiliations:** 1Rudolf Becker Laboratory for Prostate Cancer Research, Institute of Pathology, University of Bonn Medical Center, 53127 Bonn, Germany; 2MVZ Pathologie Bethesda gGmbH, Heerstrasse 219, 47053 Duisburg, Germany

**Keywords:** perineural invasion, prostate cancer, pathogenesis, prognostic significance, predictive factor

## Abstract

**Simple Summary:**

Prostate cancer is one of the most frequently diagnosed cancers in men worldwide. Perineural invasion (PNI), the movement of cancer cells along nerves, is a commonly observed approach to tumor spread and is important in both research and clinical practice of prostate cancer. However, despite many studies reporting on molecules and pathways involved in PNI, understanding its clinical relevance remains insufficient. In this review, we aim to summarize the current knowledge of mechanisms and prognostic significance of PNI in prostate cancer, which may provide new perspectives for future studies and improved treatment.

**Abstract:**

Perineural invasion (PNI) is a common indication of tumor metastasis that can be detected in multiple malignancies, including prostate cancer. In the development of PNI, tumor cells closely interact with the nerve components in the tumor microenvironment and create the perineural niche, which provides a supportive surrounding for their survival and invasion and benefits the nerve cells. Various transcription factors, cytokines, chemokines, and their related signaling pathways have been reported to be important in the progress of PNI. Nevertheless, the current understanding of the molecular mechanism of PNI is still very limited. Clinically, PNI is commonly associated with adverse clinicopathological parameters and poor outcomes for prostate cancer patients. However, whether PNI could act as an independent prognostic predictor remains controversial among studies due to inconsistent research aim and endpoint, sample type, statistical methods, and, most importantly, the definition and inclusion criteria. In this review, we provide a summary and comparison of the prognostic significance of PNI in prostate cancer based on existing literature and propose that a more standardized description of PNI would be helpful for a better understanding of its clinical relevance.

## 1. Introduction

With its high incidence, prostate cancer (PCa) is estimated to be the most commonly diagnosed cancer in men in the USA in 2022, accounting for more than one-fourth of all cancer diagnoses [1]. The development of PCa is multifactorial. Accumulating evidence has proven that, in addition to the epithelial cancer cells, the tumor stroma is also important, where investigating the roles of the neural components is an area of active research [2]. For instance, the computerized quantification and planimetry analysis of the tissue sections from PCa patients showed that the presence of tumor lesions could induce the growth of nerves, indicating complicated neuroepithelial interactions in the PCa microenvironment [3]. Meanwhile, cancer cells can also spread along nerves, a process known as perineural invasion (PNI). This is an important approach for metastasis and invasion adopted by various types of malignancies, such as colon and rectum cancer, head and neck cancer, and particularly pancreatic and prostate cancer [4,5].

Generally, PNI is a sign of severe disease and is prone to be found in more advanced tumors [6]. PCa cells within the PNI lesions show increased levels of Ki-67, EGFR, and CD74 [7]. High expression of Neuropeptide Y and positive staining of ERG also indicate more frequent PNI [8,9]. On the other hand, loss of expression of human leukocyte antigen class I (HLA-I) is associated with an increased incidence of PNI [10]. Furthermore, PCa samples with PNI are associated with increased vascular endothelial growth factor (VEGF) expression, higher microvascular density, new endoneurial capillaries formation, and morphological change of vasa nervorum with thick basement membrane and positive CD34 expression [11,12]. As for neural markers, it is common to observe upregulated neural cell adhesion molecule (N-CAM) in the nerves involved with PNI [13].

## 2. Pathogenesis of PNI in PCa

Traditional theory regarding the pathogenesis of PNI focused on the “low-resistance channel” hypothesis, which considered the perineural space as a low-resistance plane to allow tumor cells to easily spread along. However, the three layers of the nerve sheath which are composed of collagen and basement membrane are in contrast a highly resistant path [4,14]. In fact, cancer cells not only move along the nerves, but they also create the perineural niche as the microenvironment, which is beneficial for themselves, where they are found with improved proliferation, more active metabolism, and reduced apoptosis [2]. Currently, the neural tracking mechanism that cancer cells actively migrate along nerves by the support of different types of cells is becoming more widely accepted. Amit et al. proposed a model with 7 main properties to describe this complex process, of which the core concept is the interaction among various cellular components of the perineural niche [15]. This theory is also supported by accumulating data of in vitro studies. The co-culture of human PCa cells, stroma cells and mouse dorsal root ganglia improves the growth of tumor colony, outgrowth of neurite, and the presence of PNI [16,17]. Therefore, multiple cell types are involved in the pathogenesis of PNI, which may interact with each other through autocrine and paracrine mechanisms [4].

Nevertheless, the current understanding of PNI is still very limited and actively debated. For example, in the model put forward by Amit et al., the inflammation response induced by invading cancer cells also plays a critical role [15]. Consistently, using an inducible prostatitis mouse model, the POET3 mouse, Burcham et al. observed a trend of enhanced tumorigenesis after inflammation induction, including the development of PNI [18]. However, in a recent retrospective study of 1399 PCa biopsies, both acute and chronic inflammation were found to be associated with less PNI [19]. These contradictory findings underline the importance of further investigation.

## 3. Molecular Mechanisms of PNI

### 3.1. Tumor-Nerve Signaling Interaction

As discussed above, the complex tumor-nerve interaction network plays a key role in PCa progression, including the development of PNI. The pharmacological, surgical, or genetic inhibition of β-adrenergic or muscarinic receptors has been shown to prevent PCa progression [20,21]. Abnormal signaling in the glucocorticoid and serotoninergic pathways are also found associated with PNI [22]. These studies highlight the significant role of the neural components in tumor, which makes it imperative to investigate the mechanism of the emergence of the nerves in tumor. Mauffrey et al. showed that the neural progenitors from brain that express doublecortin (DCX+) could migrate to prostate tumor and initiate neurogenesis, which in turn leads to tumor growth and metastasis [23]. In addition, it has been proven that high expression of the transcription factor Snail in PCa cells enhances the adherence to nerve cells and promotes neurite outgrowth [24]. Therefore, the neurogenesis program plays an important role in the prostate tumor-nerve crosstalk. A recent single-cell sequencing study has revealed that NRXN1 and NLGN1, which are two genes encoding neuronal adhesion molecules, are closely associated with PCa progression [25]. Consistently, NLGN1 has been validated to promote PNI in PCa by cooperating with the neurotrophic factor glial cell line-derived neurotrophic factor (GDNF) [26]. Some other neurotrophic factors are reported to be involved with PNI as well, such as the nerve growth factor (NGF) and the brain-derived neurotrophic factor (BDNF) [27]. Of note, in the development of antineurogenic therapies, it is important to avoid or decrease the neuronal toxicity. Many studies focus on targeting neurotrophic factors. For example, targeting NGF and its receptors in PCa may have the effects on decreasing nerve infiltration, inhibiting tumor cell growth, and relieving the cancer-associated pain [28]. Targeting interaction between nerve and immune cells in tumor microenvironment might be another potential strategy. It is reported that the tumor-associated nerves in PCa show high expression level of programmed cell death ligand-1 (PD-L1) and inhibit the function of immune cells, which provides a new perspective for the application of immune checkpoint inhibitors [29].

N-CAM upregulation in nerves is considered a feature of PNI in PCa. Li et al. proposed the hypothesis that cancer cells induce the upregulation of N-CAM in nerves through a paracrine loop, which in turn accelerates the migration of cancer cells towards nerves through the nuclear factor kappa B (NFκB) pathway [13]. NFκB plays a critical role in various types of malignancies. In PCa, NFκB nuclear translocation has been found crucial for the inhibited apoptosis and enhanced proliferation of tumor cells in the PNI lesions. In this case, PIM2 and DAD1 may act as the downstream effectors [30,31]. However, PIM1 was not found to be significantly correlated to PNI in the immunohistochemical examination of 120 radical prostatectomy specimens, though its upregulation was observed in the adenocarcinoma tissues compared with benign epithelium [32]. Another molecule that correlates to the NFκB pathway in PNI is Semaphorin 4F (SEMA4F), which belongs to the axon guidance molecules semaphorin family. In addition to the antiapoptotic effect induced by interacting with the NFκB pathway, SEMA4F expression level has also been linked to nerve density and PNI diameter, as overexpression of SEMA4F could induce neurogenesis [33,34]. Another member of the semaphorin family, semaphorin 3C (SEMA3C), has been proven to be involved with PNI in PCa. SEMA3C can be activated by monoamine oxidase A (MAOA) in a Twist-1-dependent manner and interacts with its receptor and co-receptor PlexinA2 and neuropilin-1 (NRP1), thereby stimulating c-MET. This SEMA3C/PlexinA2/NRP1-cMET axis promotes PNI in PCa and indicates the close connections between PNI and nerve-related signals [35].

### 3.2. Chemokines, Cytokines, and Related Pathways

Plenty of studies have shown that various chemokines and cytokines play important roles in PNI. Chemokines released by nerves induce cancer cell migration and invasion towards nerves, indicating the significance of tumor-nerve interaction in PNI. For instance, CC-chemokine ligand 2 (CCL2, or MCP1), well-known as an inflammatory chemokine, has been shown to promote the progression of many cancers, including PCa [36]. The signaling axis of CCL2 and its receptor CC-chemokine receptor 2 (CCR2) facilitates PNI in PCa too. It has been reported that CCR2 expression in cancer cells accelerates PNI in the in vitro tumor-nerve co-culture invasion assays. Such effect was found to be significantly suppressed when the neurons were harvested from CCL2(−/−) knockout mice. Therefore, the nerve-released CCL2 interacts with cancer cell CCR2 to promote PNI and this CCR2-mediated signaling links to the MAPK and Akt pathway activity [37]. Many other chemokines are also reported to be highly expressed in PNI tissue. They might have close connections with the development of PNI, such as CXCL12 and its receptor CXCR4 [38], CX3CL1 and the receptor CX3CR1 [39,40], and CXCL16 [41]. However, a more detailed chemokine regulation network remains to be delineated.

Transforming growth factor-beta (TGF-β) is a pivotal cytokine in tumor biology, including PCa. It promotes PCa progression through multiple mechanisms, such as tumor cell stemness and epithelial-mesenchymal transition (EMT) [42]. A canonical pathway in the induction of EMT is the TGF-β/SMAD signaling, in which TGF-β activates SMAD2/3 to form a complex with SMAD4 and then translocates into the nucleus and acts as transcription factors [43]. Consistently, the activity of the TGF-β/SMAD pathway and the expression of one of its regulators, hexamethylene bisacetamide-inducible protein 1 (Hexim1), have also been shown to be positively correlated to PNI levels in PCa [44]. Meanwhile, Kakies et al. reported a PCa case with neuroendocrine differentiation and extensive PNI, where decreased E-cadherin and elevated vimentin and N-CAM expression were observed, indicating the potential connections among neuroendocrine differentiation, EMT, and PNI in PCa [45]. Here, both EMT and the development of neuroendocrine prostate cancer (NEPC) are prime examples of lineage plasticity in PCa, which lead to therapy resistance and poor prognosis [46]. Mucin 1 (MUC1) has been found as a robust regulator of lineage plasticity in PCa that promotes the progression to NEPC [47]. Intriguingly, MUC1 is also considered a key player in the development of both EMT and PNI [48], which supports the connection between PNI and PCa lineage plasticity. Furthermore, it has been shown that the high expression of NeuroD1, a neuronal transcription factor that facilitates neuroendocrine differentiation, is a significant risk factor for PNI. A similar effect has also been found for the neuroendocrine secretory protein Chromogranin-A [49]. In addition to the induction of EMT, TGF-β is also related to the antiapoptotic effect in PNI by forming a paracrine loop. TGF-β1 has been found to be upregulated in tumor cells that approach the nerve, which enhances the production of cav-1 in the perineurium of nerves, and cav-1 secretion, in turn, prevents tumor cells from apoptosis [50].

In the induction of EMT, the canonical TGF-β/SMAD pathway can also interact with many other signaling pathways, such as PI3K-Akt, ERK, and Wnt/β-catenin pathways [43]. Interestingly, these pathways are reported to be involved in the development of PNI in PCa as well. Overexpression of asparaginyl endopeptidase (AEP) in PCa, which activates the PI3K-Akt pathway, has been shown to be associated with more advanced tumor stage, higher Gleason score, as well as severer PNI [51,52]. Similarly, Babasaki et al. have reported that the expression of oncogene CLSPN (claspin) is positively correlated to PNI, and the knockdown of claspin inhibits Akt and ERK1/2 phosphorylation [53]. Unlike the cytoplasmic Akt, which acts as a risk factor in PCa progression, high expression of nuclear Akt isoforms Akt-1 and Akt-2 are considered to be correlated with the absence of PNI [54]. The Wnt/β-catenin pathway is also found involved with PNI in PCa. Its activation might be associated with the dysfunction of primary cilia, the microtubule-based structure which sticks out from the cell surface. Cilia are shorter in PNI foci than normal tissue, and fewer ciliated cells are observed [55]. These findings indicate that various signaling pathways tightly link tumor cells with neural components, leading to a complex regulation network of PNI (Figure 1).

### 3.3. Other Potential Mechanisms

In addition to the pathways mentioned above, many other studies focus on varied potential mechanisms to better understand PNI. Activation of the hedgehog signaling is found to be significantly correlated to PNI development [56]. In a PCa cohort with 213 patients and an average follow-up of 12 years, high expression levels of cyclin A/cyclin D are proven to be associated with several malignant characteristics, including PNI [57]. Other studies have also reported consistent results, where estrogens and the estrogen receptor beta (Erβ) are considered the potential regulator of cyclin D1 expression [58,59]. In addition, the thromboxane synthase (TxS) expression is significantly increased in PNI. Inhibition of TxS or its product thromboxane A(2) suppresses cell mobility but not cell cycle or survival [60]. Many other studies reported molecular targets which are involved with PNI level in PCa, such as beta-2 isoform of voltage-sensitive sodium channels [61], bystin [62], serine-arginine protein kinase 1 (SRPK1) [63], and PAQR3 hypermethylation [64].

Noncoding RNAs are also reported to be important players in PNI. The long noncoding RNA (lncRNA) OGFRP1 is correlated to advanced tumor stages and PNI level, which might be due to its binding with microRNA miR-124-3p, thereby preventing SARM1 from being inhibited [65]. Hu et al. reported the single-nucleotide polymorphisms (SNPs) of lncRNA H19 which might increase the risk of PNI in PCa [66]. Other noncoding RNAs such as lncRNA HOTTIP [67], microRNAs miR-224 [68], miR-301a, and miR-454 [69], are also found to be positively associated with PNI, while miR-130a is shown downregulated in tumor tissues and negatively correlated with PNI [69]. However, despite the accumulating research finding potential molecular targets and signaling pathways that might participate in the development of PNI in PCa, a thorough view of the molecular mechanism of PNI is still waiting to be further elucidated.

## 4. Prognostic Significance of PNI in Prostate Cancer

PNI is often associated with adverse clinicopathological parameters like Gleason score, prostate-specific antigen (PSA) levels, or extraprostatic extension [70,71,72]. As discussed for molecules and pathways involved in PNI, many studies also focus on the prognostic relevance of PNI. However, its significance as an independent predictive factor in PCa is still controversial. Several systematic reviews and meta-analyses suggest that PNI is, in fact, a significant prognostic factor for biochemical recurrence (BCR) of localized PCa [72,73,74]. A meta-analysis of 19 studies with a total of 13,412 patients showed a 1.4- or 1.2-fold increased risk of BCR in patients with PNI after radical prostatectomy or radiotherapy, respectively [74]. Similar results, i.e., a significant adverse association of biopsy PNI with BCR-free survival, were also reported in other meta-analyses with smaller patient cohorts [72,75]. However, systematic assessment of PNI’s clinical significance is complicated by inconsistent PNI definition and substantial variation in PNI reporting. Therefore, despite the plethora of existing studies, the overall prognostic value of PNI remains unclear. The following paragraphs will highlight some of the problems identified within PNI studies and offer suggestions to increase comparability and thus improve clinical relevance.

### 4.1. Missing Consensus Definition of PNI

The first and foremost caveat for PNI studies is the lack of a consensus definition. In 1985, Batsakis proposed the definition of PNI as tumor cell invasion “in, around, and through” the nerves [76]. With the development of the field, some researchers refined this relatively broad definition. For example, Lee et al. limited their analysis to sites where “at least one third of the nerve circumference” was involved [77]. This is in agreement with the definition advocated by Liebig et al.: “tumor in close proximity to nerve and involving at least 33% of its circumference or tumor cells within any of the 3 layers of the nerve sheath” [4]. In contrast, more stringent definitions of PNI, i.e., “presence of carcinoma in a gland that encircled an intraprostatic nerve,” are also used (e.g., Hsiang-Hsuan et al. [70]) (see Table 1). The use of different PNI definitions affects reported PNI prevalence and influences the results of these studies. This has been clearly shown in a study classifying PNI into four categories ranging from nerves with no immediate tumor cell contact to nerves fully surrounded by tumor cells. Here, only nerves with at least 50% of their circumference touched by tumor cells were significantly correlated with ISUP grade and pT categories. Kaplan-Meier survival analysis showed no significant differences when comparing PNI-negative with PNI-positive cases. Closer examination of PNI-positive cases revealed that overall survival was significantly shorter in patients with “classical PNI”, i.e., fully encircled nerves [78].

### 4.2. PNI Analysis in Biopsy vs. Radical Prostatectomy Specimens

Another source of inconsistent results lies in the variety of samples used for PNI analysis, i.e., needle biopsies, samples from transurethral resection of the prostate, or radical prostatectomy specimens. Needle biopsies generally only represent a small part of the prostate and may not include nerves, as nerves are usually only found in relatively superficial prostate biopsies [79]. Therefore, a “negative” result may indicate both the absence of PNI in a nerve-containing sample and the absence of nerve, thus bias the analysis. For example, Lee et al. found PNI in only 7.4% of patients when biopsies were investigated versus 52.1% when prostatectomy material was analyzed [77] (see also Table 1). The additional disparity may occur due to the employed biopsy technique (i.e., targeted biopsy vs. biopsies from the peripheral zone) and the number of cores/samples investigated. This is especially relevant considering that biopsy practice has dramatically changed over the past two decades, moving from, e.g., sextant anatomy-based biopsy towards MRI-guided multisite biopsies with 10–12 cores or more [80]. The effects of these changes on PNI reporting are unclear. On the one hand, in a study where the applied biopsy practice changed during the study, the rates of PNI in sextant biopsies and extended multisite biopsies were 6.3% (4/64) and 3.4% (10/297), respectively [77]. On the other hand, PNI has been shown to be significantly associated with the number of biopsy cores taken [81] and the number of positive cores [82].

Lastly, pathologists often note PNI, but it is not systematically included in pathology reports. In 2019, the International Collaboration on Cancer Reporting (ICCR) recommended but did not require PNI reporting in needle biopsy pathology records [80]. This practice, however, may skew the results of studies where PNI information is based solely on the pathological report without histologic re-examination. Compared to the original pathology report, re-evaluation of biopsies led to the detection of PNI in an additional 5.4% of cases [83].

### 4.3. Qualitative vs. Quantitative PNI Reporting

Another source for variation is PNI reporting, i.e., dichotomizing tumors into PNI-positive or PNI-negative vs. actual quantification of PNI foci. Lubig et al. show that dichotomizing samples might not be sufficient to comprehend PNI’s potential clinical relevance fully. Kaplan–Meier survival analysis showed no significant differences when comparing PNI-positive vs. -negative cases. In contrast, the number of positive nerves per high power field was significantly associated with overall survival [78]. In agreement, several studies have shown improved prognostic/predictive value using a more simplistic quantification approach. Compared with a single PNI site, multiple PNI sites were reported to be an independent predictor of disease progression [95] and associated with reduced castration-resistant PCa-free survival and overall survival [96]. A different study found that the presence of more than three PNI foci was an independent prognostic indicator for BCR [72].

PNI is usually reported based on standard hematoxylin and eosin (H&E) staining and can be quite challenging. In histological sections, nerves can have different appearances and be mistaken for bundles of stroma or smooth muscle tissue [83]. In addition, small nerves can be easily missed, leading to over- or underestimation of PNI. To improve identification of nerves, specimens can be stained for S-100 protein. Several studies have shown superior results when measuring PNI using nerve-specific S-100 stain [78,83,97].

### 4.4. Study Focus and Choice of Endpoint

Many studies are performed with curative intent and are thus biased towards specific treatments. Patients who are eligible for active surveillance will differ from those who undergo radical prostatectomy and/or radiotherapy. In addition, studies may focus on patients meeting particular inclusion criteria, e.g., patients with localized disease or patients with metastatic PCa. Distinct patient groups have inherently different baseline recurrence risks and PNI prevalence (e.g., T2 vs. T3 or low vs. high Gleason grade tumors [98]), which may add to the variations reported. In studies with heterogeneous patient cohorts, PNI may not predict progression or outcome for the entire cohort but does so for specific subgroups, e.g., low-risk patients [96,99]. Yet, as often pointed out by the authors themselves, a small sample size can be a considerable limitation of subgroup analyses. To be of clinical relevance, minimal sizes of study cohorts or subgroups should be considered. Only studies with >100 patients were included for this review, but this cutoff was set arbitrarily. Instead, cohort and subgroup size should be based on statistical power calculations as required for clinical trials.

Another reason for inconclusive results regarding the prognostic value of PNI may lie in the different choices of clinical endpoints. PCa progression is a slow process, and depending on the primary endpoint, e.g., cancer-specific death, it may take years to collect a significant number of patients. Many studies use pathological factors as surrogates for outcome and progression, e.g., biochemical (PSA) recurrence (BCR). As mentioned earlier, various studies found PNI to be an independent prognosticator of BCR (e.g., Cohn et al. [82]). Fewer studies have investigated other clinical outcomes, such as PCa-specific or overall survival. Unfortunately, even comparable studies (large patient cohorts, all treated with radiotherapy, with long follow-up periods) report conflicting results. One study shows that PNI is an independent predictor for locoregional recurrence and overall survival but not biochemical failure [100]. Opposite results are reported by Peng and colleagues, where PNI was independently associated with inferior biochemical failure-free survival but not with metastatic-free or overall survival [99]. A third study showed a prognostic value of PNI for bone metastasis but not cancer-specific survival [81]. Nonetheless, in many studies, PNI is associated with adverse outcomes (Table 2).

### 4.5. Statistical Analysis: Univariate vs. Multivariate

While many studies find PNI significantly associated with BCR or other clinical outcomes in Kaplan-Meier or univariate analyses, this significance is often lost upon multivariate analysis (Table 2). For example, in one active surveillance study, PNI was significantly associated with PCa-specific death in univariable analysis but not multivariable analysis [79]. In contrast, another study found that PNI was a significant predictor of active surveillance failure [82]. The same inconsistent results can be found in studies investigating other types of treatment. Loeb et al. investigated a cohort of patients undergoing radical prostatectomy. They found a significant relationship between PNI and various adverse pathological features as well as an increased hazard ratio for biochemical progression. After adjusting for the Gleason score, PSA level, and clinical stage, PNI failed to retain significance [85]. Similarly, in a study focusing on patients treated with external beam radiotherapy, PNI was associated with clinical stage, pretreatment PSA level, and biopsy Gleason Score as well as BCR. However, PNI remained statistically significant on both univariate and multivariate levels. Indeed, after adjusting for other clinicopathologic and treatment parameters, multivariate analysis found PNI to be a statistically significant predictor for BCR [70]. In fact, 13 of 18 studies investigated by Wu et al. showed an association of PNI with BCR status or prognosis using the Chi-square method, Log-rank test, or univariate Cox analysis. However, only six studies included multivariate analysis, and in only two, PNI remained an independent prognostic factor [106]. This has repeatedly raised the question of whether PNI is a true independent prognostic factor or merely an additional risk factor for an adverse outcome (e.g., Ramos et al. [87]).

## 5. Conclusions

Despite these caveats, many studies indicate that PNI is indeed relevant for clinical outcomes. We propose that a consistent PNI definition and reporting would facilitate more conclusive results. From a study perspective, the role of biopsy PNI could be emphasized as these samples are routinely investigated in pathology. Nerve staining and detailed quantification of PNI are too laborious for routine pathology. However, a differentiation into no PNI, single, or multiple PNI foci seems feasible and could provide more relevant information. As primary treatment decisions are often based on biopsy results, the additional PNI information may be relevant for optimal patient care. To our knowledge, there are currently no treatments directly targeting PNI in PCa. Few studies have indicated that radiotherapy may impair PNI by targeting the nerve microenvironment [107,108], but this mechanism remains to be studied in more detail. In addition, several studies have investigated the role of PNI in clinical decision-making and surgery techniques in PCa. For example, PNI was not a contraindication for bilateral nerve-sparing surgery in a cohort investigated by Loeb et al. In fact, the PNI-positive subgroup of this PCa cohort showed a significantly lower risk of progression after bilateral nerve-sparing surgery [85]. Similarly, several studies suggest that the presence of PNI should not be used to exclude patients from active surveillance but may help identify men with a higher risk of progression [79,82,87,88]. With an increasing understanding of the cellular and molecular mechanism of PNI, this information may also be used to target the pivotal molecular players to prevent PNI and may thus present a novel treatment option for PCa in the future.

## Figures and Tables

**Figure 1 cancers-14-04065-f001:**
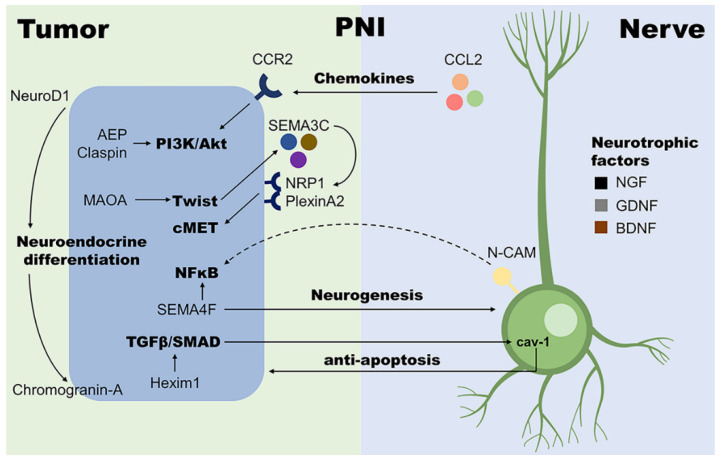
Examples of molecular mechanisms involved with PNI in PCa. With the help of nerve cells, tumor cells in PNI loci are more proliferative and invasive; on the other hand, the presence of tumor cells also facilitates neurogenesis. Such close interaction between tumor and nerve cells plays a central role in the development of PNI. AEP, asparaginyl endopeptidase; BDNF, brain-derived neurotrophic factor; CCL2, CC-chemokine ligand 2; CCR2, CC-chemokine receptor 2; GDNF, glial cell line-derived neurotrophic factor; Hexim1, hexamethylene bisacetamide-inducible protein 1; MAOA, monoamine oxidase A; N-CAM, neural cell adhesion molecule; NFκB, nuclear factor kappa B; NGF, nerve growth factor; NRP1, neuropilin-1; SEMA3C, semaphorin 3C; SEMA4F, semaphorin 4F; TGF-β, transforming growth factor-beta.

**Table 1 cancers-14-04065-t001:** Examples of studies using different PNI definitions and sample types and their reported PNI rates.

Reference	PNI Definition	Sample Type	PNI Positivity *
Ahmad et al., 2018 [79]	Cancer tracking along or around a nerve within the perineural space	Biopsy	288/988 (29.1%)
Celik et al., 2018 [84]	Extension of PCa cells along the nerve bundle	Biopsy	87/380 (22.9%)
Cohn et al., 2014 [82]	Adenocarcinomas observed within the perineural space adjacent to an intraprostatic nerve fiber	Biopsy	14/165 (8.5%)
Hsiang-Hsuan et al., 2007 [70]	Presence of carcinoma in a gland that encircles an intraprostatic nerve	Biopsy	112/586 (19.1%)
Loeb et al., 2010 [85]	PCa extension along the perineural sheath	Biopsy	188/1256 (15.0%)
Merrick et al., 2005 [86]	Carcinoma tracking along, or around, a nerve within the perineural space	Biopsy	133/512 (26.0%)
Ramos et al., 2020 [87]	Circumferential or longitudinal tracking of PCa cell along a nerve, within the perineural space	Biopsy	57/107 (53.3%)
Saeter et al., 2015 [88]	Growth of cancer in the surrounding perineural space of nerves	Biopsy	141/281 (50.2%)
Ström et al., 2020 [89]	Prostatic carcinoma found immediately adjacent to a nerve, either along the nerve or surrounding it	Biopsy	146/918 (15.9%)
Lee et al., 2010 [77]	Tumor cells within any layer of the nerve sheath or tumor cells in the perineural space that involves at least one-third of the nerve circumference	Biopsy Prostatectomy	14/361 (3.9%) 188/361 (52.1%)
Lian et al., 2020 [90]	Trajectory of tumor cells along or around nerve fibers	Prostatectomy	127/416 (30.5%)
Maru et al., 2001 [91]	Carcinoma within the perineural space adjacent to a nerve	Prostatectomy	477/640 (75%)
Masieri et al., 2009 [92]	Adenocarcinoma within the perineural space adjacent to a nerve; focal contact between the tumor and a nerve was disregarded	Prostatectomy	157/239 (65.7%)
Kraus et al., 2019 [93]	Infiltration of cancer cells into the perineural space where they track along or around a nerve	Prostatectomy ^+^	936/1549 (60.4%)
Özcan et al., 2001 [94]	Adenocarcinoma glands in the perineural space within the prostate tissue	Prostatectomy ^+^	61/178 (34.3%)
Wu et al., 2020 [72]	PCa infiltration in any layer of the nerve sheath or tumor invasion involved at least one-third of nerve circumference	Prostatectomy	530/721 (73.5%)

* PNI positivity: number of patients with PNI relative to all patients included in the final analysis. Top and bottom parts of the table list studies investigating biopsy and prostatectomy specimens, respectively. ^+^ based on pathology report; PCa: prostate cancer; N/A: not available.

**Table 2 cancers-14-04065-t002:** Examples of studies investigating the clinical significance of PNI for different endpoints.

Reference	Mean/Median Follow Up in Years	Endpoint	No. of Patients Reaching Endpoint (% of Cohort)	Prognostic Significance of PNI for Endpoint (Univariate Analysis)	PNI as Independent Predictor for Endpoint (Multivariate Analysis)
Cohn et al., 2014 [82]	-/0.5	AS failure	40 (24.2%)	Yes	Yes
De la Taille et al., 1999 [101]	2.1/-	BCR	46 (14.4%)	Yes	Yes
Katz et al., 2013 [102]	4.3/3.9	BCR	56 (19.6%	Yes	N/A
Kraus et al., 2019 [93]	-/2.2	BCR	96 (6.2)	Yes	No
Lee et al., 2010 [77]	3.5/-	BCR	83 (23.0%)	No	-
Lian et al., 2020 [90]	-/2.3	BCR	94 (22.6%)	Yes	No
Loeb et al., 2010 [85]	2.8/-	BCR	57 (4.5%)	Yes	No
Masieri et al., 2009 [92]	5.5/5.2	BCR	11 (4.6%)	No	-
Merrilees et al., 2008 [97]	2.4/2.2	BCR	27 (25.7%)	No	-
Ramos et al., 2020 [87]	5.9/-	BCR	31 (29.0%)	Yes	No
Andersen et al., 1999 [103]	-/4.0	bNED	35 (12.2%)	Yes	Yes
Bonin et al., 1997 [104]	2.4/2.3	bNED	109 (22.5%) ^+^	Yes	No
Delahunt et al., 2020 [81]	-/10.6	Bone Met	212 (21.7%)	Yes	Yes
		Soft Tissue Met	171 (17.5%)	Yes	No
		DOD	130 (13.3%)	Yes	No
		Death	344 (35.3%)	Yes	No
Tollefson et al., 2014 [105]	-/12.9	Cancer Progression	135 (29.9%)	Yes	Yes
		Local or systemic progression	46 (10.2%)	N/A	Yes
		DOD	18 (4.1%)	Yes	Yes
Ahmad et al., 2018 [79]	-/9.7	DOD	169 (17.1%)	No	-
Saeter et al., 2015 [88]	-/9.2	DOD	58 (20.6%)	Yes	Yes/No (dependent on model)
Peng et al., 2018 [99]	-/11.3	RFS	N/A	Yes	Yes
		MTS	N/A	Yes	No
		CSS	74 (8.3%)	Yes	No
		OS	368 (41.4%)	No	No

^+^ numbers not provided/approximated based on study info. AS: active surveillance; bNED: biochemical no evidence of disease; BCR: biochemical recurrence (PSA-based); DOD: death of disease/PCa-specific death; DFS: disease-free survival; CSS: cancer-specific survival; OS: overall survival; RFS: recurrence-free survival; MTS: metastasis-free survival; N/A: not available.

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
