# Peer review of "The Role of Perineural Invasion in Prostate Cancer and Its Prognostic Significance"

_cancers, 2022, doi:10.3390/cancers14174065_

Round 1

Reviewer 1 Report

In the current review, authors have done a great job in describing perineural invasion (PNI) of prostate cancer and summarizing the current knowledge on multiple aspects of PNI. The review summarized and discussed the Pathogenesis, Molecular Mechanisms, and prognostic significance of PNI in prostate cancer. The figure and the tables are also very informative. There is only one concern, can authors discuss any current treatment strategies specifically targeting PNI or any prostate cancer treatment will attenuate PNI?

Author Response

We thank the reviewer for the valuable comment. To our knowledge, there are currently no treatments directly targeting PNI in prostate cancer. Some studies reported strategies targeting nerve-tumor crosstalk to inhibit tumor innervation and cancer progression, but these antineurogenic therapies warrant further investigation for clinical application. Detailed discussion regarding this issue can be found in line 107-115 and line 383-393 in the revised manuscipt.

Reviewer 2 Report

Anti-androgens are the mainstay of prostate cancer therapy and with the development of more potent drugs inhibiting androgen receptor (AR) signaling; the risk of emergence of treatment resistant prostate tumors has been increased which shows more lineage plasticity and leads to poor prognosis. It has been well established that anti-androgen therapy pushed the prostate tumor cells towards more basal like phenotype which are enriched in stem cell gene signature and difficult to target therapeutically. The most aggressive form of CRPC after anti-AR therapy is t-NEPC (treatment induced NEPC) which is not very well characterized and a hot topic of research. Clinically, lineage plasticity may be characterized as AR-/lo and PSA-/lo which makes anti-AR therapy ineffective and sometimes small cell or neuroendocrine features. Currently, there is no approved therapeutic regimen for patients with NEPC due to their high mutation burden, activation of developmental pathways and gap in knowledge. Most of the studies are conducted on either adenocarcinoma when majority of the tumor cells are ARhi or end point NEPC tumors and nothing much is know about the events happening in between.

This review article is a very well written explaining the emergence of neuronal phenotype in prostate tumors and how this phenomenon facilitates the crosstalk between tumor cells and neurons. However, there are some suggestions to improve this article further-

1.       Authors should include a section mentioning the role of neurogenesis program which involves NRX1, NLGN1, type of proteins in this prostate tumor-neuronal crosstalk.

2.       It will be of great interest to other researchers in the field if authors discuss- How this whole PNI connected to linage plasticity in prostate tumors.

3.       Authors missed a key reference (Mauffrey et al., Nature. 569:672) in this article and should be discussed in detail.

4.       One problem with targeting this tumor-neural crosstalk is the potential side-effects on brain and nervous system. Authors may include a section discussing the role of factors (which may include master regulators like transcription factors) which express exclusively in tumor cells.

Author Response

Comment 1. Authors should include a section mentioning the role of neurogenesis program which involves NRX1, NLGN1, type of proteins in this prostate tumor-neuronal crosstalk.

Answer: We thank the reviewer for this important suggestion. The neurogenesis program and related molecules are important in prostate cancer PNI. A section including the discussion of NRXN1, NLGN1, and other factors has been added in line 97-104 in the revised manuscript.

Comment 2. It will be of great interest to other researchers in the field if authors discuss- How this whole PNI connected to linage plasticity in prostate tumors.

Answer: We thank the reviewer for the valuable advice. It is interesting to include this issue in our manuscript. In the original manuscript we mentioned a study reporting the potential connections among neuroendocrine differentiation, EMT, and PNI in prostate cancer. In the revised manuscript, more detailed discussion regarding PNI and lineage plasticity in prostate cancer has been added, which can be found in line 162-173.

Comment 3. Authors missed a key reference (Mauffrey et al., Nature. 569:672) in this article and should be discussed in detail.

Answer: We are sorry for our negligence. This is a very important reference for our article and has been discussed in detail in the revised manuscript, which can be found in line 93-97.

Comment 4. One problem with targeting this tumor-neural crosstalk is the potential side-effects on brain and nervous system. Authors may include a section discussing the role of factors (which may include master regulators like transcription factors) which express exclusively in tumor cells.

Answer: We thank the reviewer for this valuable suggestion. A paragraph discussing antineurogenic therapies has been added in the revised manuscipt, which can be found in line 107-115.
